# The Impact of Mechanized and Traditional Processes on Microbial Diversity and Volatile Flavor Compound Formation During Xifeng Baijiu Fermentation

**DOI:** 10.3390/foods13223710

**Published:** 2024-11-20

**Authors:** Chengyong Jin, Guangyuan Jin, Juan Jin, Yutao Lv, Zhe Dang, Yafang Feng, Yan Xu

**Affiliations:** 1The Lab of Brewing Microbiology and Applied Enzymology, School of Biotechnology, Jiangnan University, Wuxi 214122, China; jincy24@163.com (C.J.); g.jin@jiangnan.edu.cn (G.J.); 2Shaanxi Xifeng Liquor Co., Ltd., Baoji 721406, China; m13369494215@163.com (J.J.); 18629277662@163.com (Y.L.); 17791059691@163.com (Z.D.); 13892788028@163.com (Y.F.)

**Keywords:** Xifeng Baijiu, traditional process, mechanized process, solid-state fermentation, microbial diversity, volatile flavor compounds

## Abstract

The impact of mechanized processes on the properties of Xifeng Baijiu, as well as the differences between Baijiu produced through mechanized versus traditional methods, remains insufficiently understood. In this study, the differences in physicochemical properties, microorganisms, volatile flavor compounds, and their correlations in the traditional and mechanized processes of producing Xifeng Baijiu were compared. High-throughput sequencing revealed that the abundance and diversity of bacteria and fungi were higher in the traditional process compared to the mechanized one. The bacterial population exhibited a more pronounced succession pattern than the fungal population throughout the fermentation. In the early stages, Firmicutes and Actinobacteria were the dominant bacterial phyla in both processes, with *Lactobacillus*, *Saccharopolyspora*, *Bacillus*, *Acetobacter*, *Weissella*, and *Thermoactinomyces* being the predominant bacterial genera, and *Saccharomycopsis*, *Issatchenkia*, *Kazachstania*, *Thermoascus*, *Pichia*, and *Rhizopus* are the dominant fungi. Chemical analysis identified 71 volatile flavor components in the fermented grains, predominantly esters and alcohols. Ethyl caproate, 1-nonyl alcohol, ethyl acetate, acetic acid, butyric acid, furfuryl alcohol, caproic acid, and 2,4-di-tert-butylphenol were the key differential compounds between the two production methods. Pearson correlation analysis indicated a stronger relationship between bacteria and flavor compounds than between fungi and these substances, with *Lactobacillus* showing a negative correlation with other dominant bacterial genera. These findings offer a foundation for future research into the factors contributing to differences in Baijiu produced by traditional and mechanized methods and serve as a reference for improving mechanized processes.

## 1. Introduction

Baijiu, a traditional alcoholic beverage made from grains using solid-state fermentation and distillation [1,2], plays an important role in cultural, economic, and social activities [3,4]. Xifeng Baijiu, a famous Chinese Baijiu, has special characteristics owing to its unique brewing technology. Pure grain brewing coupled with sea-stored wine provides Xifeng Baijiu with mellow, elegant, sweet, and cool flavors.

In recent years, with the rapid development of microbiome technology, microorganisms in Baijiu-brewing environments have been extensively studied. For example, fungi play a key role in solid-state fermentation [5]. They can produce amylase, thereby hydrolyzing the starch in raw sorghum into sugars that can directly participate in fermentation. *Issatchenkia*, *Talaromyces*, *Aspergillus*, and *Eurotium* are the dominant fungal genera in the fermentation of sauce-flavor Baijiu in China [6]. The assembly and succession of microbial communities in the second round of fermentation of light-flavor Baijiu were studied. It found that *Streptomyces*, *Bacillus*, and *Lactobacillus* were the dominant bacteria in the early stages of fermentation, whereas *Lactobacillus* was the dominant bacterium in the middle and late stages of fermentation [7]. In the Baijiu-brewing process, microorganisms are found throughout the fermentation system and interact and influence each other, forming a complex brewing micro-ecology. Therefore, the fermentation process is essentially a “multi-micro-fermentation” process. Microorganism contents in the brewing environment of Xifeng Baijiu, the fermented grains, and Daqu have been studied, but investigations of the principles and mechanisms related to microbial composition are required.

Simultaneously, Baijiu’s aroma components have gained research interest. Baijiu is mainly composed of ethanol, water, and trace components, of which water and ethanol account for approximately 98–99% of the contents, and trace components, including esters, acids, alcohols, aldehydes, and ketones, account for only 1–2%. Although the content of trace components is low, as important aromatic substances in Baijiu, their content and composition determine the characteristics of Baijiu. For Xifeng Baijiu, the volatile flavor substance contents in Baijiu with different storage durations and container types have been identified using a variety of methods, such as headspace solid-phase microextraction gas chromatography-mass spectrometry and comprehensive two-dimensional gas chromatography-time of flight mass spectrometry, and the key volatile flavor substances in Xifeng Baijiu have been determined [8,9]. However, further research is needed on the characteristic aroma substances and vintage signal substances of Xifeng Baijiu.

Flavor is the main characteristic of fermented products [10] such as Baijiu. The type and quality of Baijiu depend on its volatile substance contents, and microorganisms are closely related to the formation of volatile substances. Increasingly, studies on the correlation between microorganisms and flavor substances are being conducted. For example, *Acinetobacter*, yeast, and other microorganisms were significantly correlated with the contents of n-propanol, 2,3-butanediol, and other components of the base wine [11]. *Lactobacillus* was positively correlated with ester content [12]. These results have significant implications for the development of Baijiu with certain flavors and directional control of microorganisms during fermentation.

With the progress of science and technology and the development of society, the disadvantages of the traditional brewing process of Chinese Baijiu, including extensive processing, low production efficiency, low standardization, and a low degree of refinement, have been highlighted. Therefore, mechanized Baijiu-brewing methods have been developed, and research on the mechanization of Baijiu brewing has gradually increased. *Bacillus*, *Lactobacillus*, and *Sphingomonas* play important roles in the biological regulation of the core microbial communities in grains in different rounds of sauce-flavor Baijiu fermentation [13,14,15]. A study on microbial community evolution during manual and mechanized brewing of Xiaoqu Qing-flavor Baijiu showed that *Lactobacillus* was the most abundant bacterium in both manual and mechanized brewing. The abundance of *Lactobacillus* in mechanized brewing was higher than in manual brewing, and the fungal diversity in manual brewing was higher than 1in mechanized brewing [16]. Existing research on the Xifeng Baijiu brewing workshop mainly focuses on some microorganisms in the brewing workshop environment [17,18], while a comparison of the whole microenvironment throughout the fermentation processes of traditional and mechanized workshops has not been conducted.

In this study, the microorganisms present in the environment and fermentation processes of mechanized and traditional workshops were investigated. The bacterial and fungal community compositions during fermentation using the two brewing methods were compared to determine the differences caused by the different environments. The effect of a mechanized brewing method on environmental microorganisms was also investigated. Moreover, the content of volatile flavor compounds during the fermentation process was detected to further determine the relationship between microbial communities and metabolites. Finally, we traced the sources of fermentation flora and evaluated the impact of the environment on fermentation flora abundance. This study provides a basis for understanding the brewing process of Xifeng Baijiu from the perspective of environmental microorganisms, which can support the improvement in the quality of Baijiu and promote the advancement of the Baijiu industry.

## 2. Materials and Methods

### 2.1. Experimental Design and Sample Collection

The fermented grains were collected from both traditional and mechanized workshops in Shaanxi Xifeng Liquor Co., Ltd., Baoji, China (107.325677° E, 34.549184° N). The traditional and mechanized workshops were two separate workshops built side by side, with the same source, transport, and storage conditions for the grains used for brewing, and the same brewing process. In addition, the Daqu and water used for brewing in the two different types of workshops were also the same. The difference was that the entire production process of the mechanized workshop was completed by machine, and the lids of the pits were made of cement, while the production process of the traditional workshop was completed manually and the lids of the pits were made of stainless steel. In addition to the above differences, the traditional workshop has been in production for a longer time, while the mechanized workshop has been in production for a shorter time.

Three pits in the mechanized and traditional workshops were selected for sampling. Samples of fermented grains in the top, middle, and bottom layers were collected by a customized sampler similar to a soil drill at eight different time points (day 0, day 4, day 7, day 10, day 14, day 18, day 22, and day 30) during the round-cellar period. For each layer, the fermented grains were collected at three points: the central position and the two points near the two ends of the pit wall (Figure 1). Subsequently, the samples were mixed and stored in a Ziplock bag, and Daqu and environmental samples (from the water, tools, ground, and air) were collected simultaneously. Then, the collected samples were stored at −80 °C.

### 2.2. Determination of Physical and Chemical Indices

Determination of the temperature of the fermented grains: a thermometer was inserted near the sampling point and left for approximately 1 min while the sample was collected. After the thermometer readings stabilized, the temperature data were recorded.

Determination of moisture, acidity, starch, and reducing sugar in fermented grains: near-infrared spectral analysis was conducted using the Antaris II FT-NIR Analyzer (Thermo Fisher Scientific Inc., Walthan, MA, USA) to measure moisture, acidity, starch, and reducing sugar in fermented grains [19].

### 2.3. DNA Extraction

DNA was extracted from all samples using the indirect extraction method according to the instructions of the FastDNA^®^ soil DNA extraction kit. Then, the quality of the extracted DNA was evaluated using an enzyme-labeled instrument, and the content of the extracted DNA was determined by electrophoresis using a 1% agarose gel. The DNA samples that met the standards were immediately stored in a freezer at −80 °C [20].

### 2.4. PCR Amplification and High-Throughput Sequencing

The primers F338 (5′-ACTCCTACGGGAGGCAGCAG-3′) and R806 (5′-GGACTACHVGGGTWTCTAAT-3′) were used to amplify the bacterial 16S rRNA V3-V4 region gene sequence. The gene sequence of the fungal ITS region was amplified using the F1737 (5′-GGAAGTAAAAGTCGTAACAAGG-3′) and R2043 (5′-GCTGCGTTCTTCATCGATGC-3′) primers. The amplified products were recovered, quantified, and sequenced on an Illumina MiSeq platform, and a database was constructed according to the required data depth. High-throughput sequencing was performed by Meiji Biomedical Technology Co., Ltd (Shanghai, China).

### 2.5. High-Throughput Sequencing Data Analysis

After the PE reads obtained by sequencing were split, the double-ended reads were quality-checked and filtered according to sequencing quality. Double-ended reads were spliced based on any overlap to optimize the data after quality control splicing. Then, the DADA2 sequence denoising method was used to process the optimized data to obtain the amplicon sequence variant (ASV) representing the sequence and abundance information. Based on the ASV representing sequence and abundance information, taxonomic, community diversity, species difference, correlation analyses, and statistical and visual analyses were performed.

### 2.6. Determination of Volatile Flavor Substances

Ultrapure water (40 mL) was added to 10 g of fermented grains and ultrasonicated in an ice bath for 30 min with ultrasonic equipment (SB-800DTD, SCIENTZ, Zhejiang, China). The mixture was centrifuged at 4 °C at 10,000 r/min for 20 min using a refrigerated centrifuge (GL-21M, cence, Changsha, China). Then, 8 mL supernatant was extracted into a 20 mL headspace bottle, and 3 g NaCl and 20 μL internal standard (geranyl acetate) were added to a gas chromatograph mass spectrometer (GCMS-QP2020NX, Shimadzu Company, Kyoto, Japan) for detection. The gas chromatography-mass spectrometry conditions were as follows. The column was a Shim-pack GIST C18 column (Shimadzu Company, Kyoto, Japan; 4.6 mm × 250 mm, 5 μm), the column temperature was 30 °C, the detection wavelength was 210 nm, the flow rate was 1 mL/min, the mobile phase A was methanol, and B was 0.02 mol/L potassium dihydrogen phosphate solution. Heating procedure: the solution was heated at 50 °C for 2 min, followed by 4 °C/min temperature increases up to 230 °C, which was maintained for 15 min. The temperature of the inlet and the detector was 250 °C, the carrier gas was high-purity nitrogen, the flow rate was 1 mL/min, the shunt ratio was 37:1, the tail-blowing air was 20 mL/min, the airflow rate was 400 mL/min, and the hydrogen flow rate was 40 mL/min [9].

### 2.7. Data Analysis

Excel 2010 was used for statistical analysis, and IBM SPSS (version 26.0) was used to perform analysis of variance (ANOVA) and Duncan’s test; differences were considered statistically significant at *p* < 0.05. Data were plotted using the Origin 2023 software. Statistical analyses of all sequencing data were performed in R (Version 4.3.1). The Bray–Curtis distance between samples was calculated using the “Vegan” package in R, and principal component analysis (PCA) of bacteria and fungi in the fermentation process of fermented grains was conducted through the “Vegan” package operation. Linear discriminant analysis effect size (LEfSe) was calculated using the “microeco” package in R. Canonical correspondence analysis (CCA) was performed to analyze the relationship between microorganisms and metabolites during fermentation. In addition, correlations between variables were examined using the Pearson correlation coefficient and were visualized using the “ggplot2” package in R.

To explore the source of microorganisms in fermented grains, traceability analysis was conducted using SourceTracker and default parameters. The grains before fermentation were set as the “sink”, and potential microbial sources such as the ground, air, Daqu, tools, and water were set as the “source”.

## 3. Results and Discussion

### 3.1. Changes in Physicochemical Indices During Fermentation

With the extension of fermentation time, the moisture content of the fermented grains from the two different types of workshops generally increased and was the same at the late stage of fermentation (Figure 2a). This indicates that the starch in the raw materials was rapidly hydrolyzed to produce ethanol and water in the early stage of fermentation, which increased the water content of the fermented grains, whereas in the late stage of fermentation, the water content of the fermented grains stabilized.

Water is a crucial factor in the Baijiu-brewing process, as it is required in every stage of the fermentation process and affects the growth, reproduction, and metabolic activities of microorganisms. During the first four days of fermentation, the acidity of the fermented grains from the two different types of workshops increased greatly and then gradually stabilized; therefore, days 0 to 4 were the main acid-production stage of grain fermentation from the two different types of workshops (Figure 2b). In addition, owing to the tools, operating environment, and personnel, the brewing microenvironment is complex, resulting in differences in acidity between the grains from the mechanized and traditional workshops. Acids are important substances for the synthesis of Baijiu flavor substances [21]. A suitable acidity level can inhibit the growth of pathogenic microbes and support the growth of yeast and other beneficial microorganisms. The content of reducing sugar and starch in the fermented grains decreased with increasing fermentation time (Figure 2c,d). Starch consumption was rapid in days 0 to 4 of fermentation; this might be due to the high growth and propagation of yeast in the early fermentation period, which requires a large amount of matter and energy, resulting in high starch consumption [22].

However, because solid-state fermentation of Baijiu is a natural multi-strain fermentation process, the differences in the content of water, acidity, starch, and reducing sugars in the fermented grains from the two different types of workshops may be caused by differences in microflora in the two different types of workshops.

### 3.2. Changes in the Content of Volatile Flavor Compounds During Fermentation

Flavor substances in the fermented grains from traditional and mechanized workshops were detected using GC-MS. A total of 71 volatile components were detected, including 41 esters, 13 alcohols, 5 acids, 3 phenols, and 9 aldehydes and ketones (Figure 3a).

Esters are the main cause of Baijiu aromas, and different esters produce different types of aromas. For instance, ethyl esters mainly produce floral and tropical fruit aromas in Baijiu [23]. Alcohols constitute a key part of the flavor backbone of Baijiu and could react with acids to produce aromatic substances, reduce the spiciness of Baijiu, and increase the harmony and fullness of the Baijiu [24]. Although the content of alcohols in the Baijiu is not high, it also contributes to the flavor of the Baijiu. Comparative analysis showed that the content of esters and alcohols in the fermented grains from the two different types of workshops was higher than other components (Figure 3b), which was consistent with the findings of many previous studies on the content of volatile flavor substances in fermented grains [22,25]. There was no difference in the esters and alcohols between the grains from the two different types of workshops, but the content of acids was higher in the grains from the traditional workshop compared with that from the mechanized workshop, which may be due to the production environment formed in the traditional workshop over a long time. Compared with the mechanized workshop, the traditional workshop has longer production lifespans and involves more human activities in the production process, which leads to different microbial environments in the two types of workshops, and ultimately affects the composition of the fermented grains.

### 3.3. Screening of Key Metabolites and Volatile Flavor Compounds During Fermentation

The flavor substances in the fermented grains from the traditional and mechanized workshops were significantly separated in the PCA, with the first principal component (PC1) at 58.82%, second principal component (PC2) at 12.84%, and total contribution degree at 71.66% (Figure 3d). Based on the results of the quantitative analysis of volatile compounds, partial least squares discriminant analysis (PLS-DA) was performed to determine volatile compound concentrations in different samples, and a VIP score chart was obtained. The results showed eight volatile compounds with a VIP > 1: ethyl hexanoate, 1-nonyl alcohol, ethyl acetate, acetic acid, butyric acid, furfuryl alcohol, hexanoic acid, and 2,4-di-tert-butylphenol (Figure 3c). These compounds were differential metabolites that could distinguish the mechanized and traditionally fermented grains. The VIP values of 2,4-di-tert-butylphenol and hexanoic acid were the highest, indicating that these two substances played a very important role in distinguishing the grains fermented using different brewing methods. Hexanoic acid was the predominant volatile compound in strong-flavor Baijiu, Nong-flavor and Jiang-flavor Baijiu, special-flavor Baijiu, Feng-flavor Baijiu, Fuyu-flavor Baijiu, Chinese medicinal Baijiu, and sesame-flavor Baijiu. When hexanoic acid was absent in strong-flavor Baijiu, the “jiao-aroma” of the Baijiu body was weakened [26]. In addition, 2,4-ditert-butylphenol significantly contributed to the burned taste of the Baijiu and could interact with the skeletal components in the base of the Baijiu to characterize the burned taste within a certain concentration range. However, when the concentration range was exceeded, it inhibited the characterization of the burned taste [27].

### 3.4. Analysis of Microbial Diversity During the Fermentation of Grains

#### 3.4.1. Analysis of Microbial α Diversity in Fermented Grains

The dilution curves of bacteria and fungi in all samples gradually increased with the increase in sequencing depth and eventually tended to plateau (Figure A1), indicating that the sequencing data were reasonable in volume, covered the population information of most microorganisms in the samples with high reliability, and could be used for subsequent bioinformatics analyses.

The Chao index was used to characterize the abundance of bacteria and fungi in the fermented grains, and the Shannon index was used to evaluate the diversity of bacteria and fungi. The Chao index of the fermented grains from the two different types of workshops decreased gradually with fermentation time, and that of the traditionally fermented grains was higher than that of the mechanically fermented grains. The bacterial Shannon index of the grains from the two different types of workshops increased and then decreased with fermentation time, and that of the fermented grains from the traditional workshop was higher than that of those from the mechanized workshop (Figure 4a,b).

In contrast to the bacterial indices, the Chao and Shannon indices of the fungi in the grains from both workshops increased with fermentation time, and those of the grains from the traditional workshop were higher than those from the mechanized workshop (Figure 4c,d). These results showed that the abundance and diversity of bacteria and fungi in the fermented grains from the traditional workshop were higher than those in the fermented grains from the mechanized workshop. The prominence of bacteria gradually decreased, and fungi gradually became the dominant microorganisms during the fermentation process.

The Venn diagram was used to analyze the differences in microbial community composition between grains at different stages of fermentation. The number of bacterial ASVs in fermented grains in the traditional workshop reached a maximum value (977) at day 4 and then decreased, reaching a minimum value of 148 at day 0 (Figure A2a), while the number of fungal ASVs reached a maximum value of 39 at day 30 (Figure A2b). The grains from the mechanized workshop had the highest number of bacterial ASVs (1199) and the lowest number of fungal ASVs (8) at day 0, and the number of bacterial ASVs was the lowest and the number of fungal ASVs was the highest at day 30 (Figure A2c,d). This was similar to the results of the diversity analysis described above. This may be because, as fermentation progressed, the oxygen content and pH in the pit decreased, and the microenvironment of fermented grains became unfavorable for bacterial growth, while the fungi occupied the bacterial ecological niche and gradually became the predominant fermentation population [28].

#### 3.4.2. Analysis of Microbial β Diversity in Fermented Grains

The types of flavor substances in the fermented grains from the traditional and mechanized workshops were significantly different. To study the β diversity of microbial communities in fermented grains from the two different types of workshops and different fermentation times, the differences in community structure were visualized using PCA of Bray–Curtis distances based on the numbers of ASVs.

There was a significant difference in the bacterial populations between the grains from the traditional and mechanized workshops (*p* = 0.001), which increased at the end of fermentation (day 30) (Figure A3a,b). This may be related to environmental differences in the pits of the two different types of workshops. In the traditional workshop, bacterial populations in fermented grains showed an obvious succession trend with the progression of fermentation, and day 14 was used as the time cut-off to divide the bacterial communities into two groups, which exhibited significant differences: the early phase of fermentation and the middle and late phases of fermentation (Figure 5a). Similar succession patterns were observed in the grains from the mechanized workshop, but they were not as significant as those in the traditional workshop (Figure 5b).

The fungal populations exhibited no significant differences between the grains from the traditional and mechanized workshops, although the populations of fungi in the fermented grains from the two different types of workshops at the end of fermentation (day 30) were slightly dispersed compared with those at the beginning of fermentation (day 0) (Figure A3c,d). With the progression of fermentation, the fungal populations in later fermentation periods were clustered, showing no obvious succession trend (Figure 5c,d), indicating that fungal populations played a key role throughout the fermentation process, and multiple strains interacted to promote fermentation.

According to these results, the difference in the microbial communities between the grains from the two different types of workshops at the end of fermentation was more significant than that at the beginning of fermentation, and the bacterial population had a more significant succession rule than the fungi throughout the fermentation process. We propose that there were differences in the working environments and methods of sealing the pit between the two different types of workshops, which would affect the growth and propagation of microorganisms in the fermentation process and ultimately lead to the different compositions of microbial communities in the fermented grains from the two different types of workshops. As fermentation progressed, the bacterial population responded more strongly to the unfavorable environment of the pit and thus showed an obvious succession rule.

### 3.5. Succession Trends of Microbial Communities

#### 3.5.1. Changes in Bacterial Community Structures

Firmicutes, Proteobacteria, and Actinobacteria were the dominant bacterial phyla, with an average relative abundance of >1% in the fermented grains from the traditional and mechanized workshops. With the progression of fermentation, the relative abundance of Firmicutes in the grains of the traditional and mechanized workshops gradually increased, ranging from 53.85% to 99.69% and from 44.78% to 99.47%, respectively, whereas the relative abundances of Actinobacteria and Proteobacteria showed a decreasing trend. During the same fermentation period, the relative abundance of Proteobacteria in the grains from the traditional workshop remained higher than that in the grains from the mechanized workshop, whereas the relative abundance of Actinobacteria in the grains from the mechanized workshop remained higher than that in those from the traditional workshop, except for on day 10 (Figure A4a,b). Firmicutes were the dominant bacterial phylum during the fermentation of light-flavor Baijiu [22,29]. In addition, these dominant bacteria were also the dominant bacteria in the pit mud during the brewing process of feng-flavor Baijiu and some other flavors, such as sauce-flavor Baijiu [30]. Firmicutes, Proteobacteria, and Actinobacteria were the predominant bacterial groups, with an abundance of >1% during fermentation, and as fermentation progressed, Firmicutes became the dominant bacterial phylum [31].

The dominant bacterial genera with an average relative abundance >1% in the fermented grains were *Lactobacillus*, *Saccharopolyspora*, *Bacillus*, *norank_f__Pseudonocardiaceae*, *Streptomyces*, *Acetobacter*, *Pediococcus*, *Staphylococcus*, *Weissella*, *Thermoactinomyces*, *Achromobacter*, *Kroppenstedtia*, and *Leuconostoc*. The relative abundance of *Lactobacillus* in the grains from the traditional and mechanized workshops gradually increased with the extension of the duration of fermentation (20.27–98.44%, 15.16–98.7%), and *Lactobacillus* became the dominant bacterial genus in the fermentation process of the two different types of workshops after day 7. The relative abundance of *Saccharopolyspora* decreased gradually with increasing fermentation time (Figure 6a,b). *Lactobacillus*, as the dominant genus of bacteria in fermented grains used for Feng-flavor Baijiu, can produce lactic acid and is positively correlated with ethyl lactate content [32]. Similar findings were also reported for sauce-flavor Baijiu, in which the abundance of *Lactobacillus* increased significantly during fermentation and persisted until the end of fermentation reaching up to 60% in the middle and late fermentation stages [13].

#### 3.5.2. Changes in Fungal Community Composition

Ascomycota and Mucoromycota were the predominant fungal phyla with an average relative abundance of >1% in the fermented grains from the traditional and mechanized workshops. Among them, Ascomycota was dominant, with relative abundances of 94% and 98% in the grains from the traditional and mechanized workshops, respectively (Figure A4c,d). Previous studies found that Ascomycota was the main fungi in the fermented grains of strong-flavor, sauce-flavor, and light-flavor Baijiu [22]. The most common fungal genera with relative abundance > 1% in the fermented grains of the traditional and mechanized workshop were Saccharomycopsis, Issatchenkia, Kazachstania, Thermoascus, Naumovozyma, unclassified_f__Aspergillaceae, Pichia, Rhizopus, and Candida. During the same fermentation period, the relative abundances of Kazachstania and Naumovozyma in the grains from the mechanized workshop were higher than those in the traditional workshop, and they comprised a certain proportion of the community from day 7, indicating that they mainly played a role in the middle and late fermentation stages. The relative abundance of the Saccharomycopsis in grains from both types of workshops decreased throughout fermentation, from 75.22% to 25.91% and 80.32% to 19.97%, respectively. In contrast to those of bacteria, the fermentation modes of fungi gradually evolved from Saccharomycopsis to Issatchenkia, Kazachstania, Thermoascus, Pichia, and other fungal co-fermentation modes. Saccharomycopsis can provide hydrolases, such as amylase and glucoamylase, for fermentation; its relative abundance is high in the early fermentation period but decreases in the middle and late fermentation periods (Figure 6c,d). Rhizopus, Pichia, and other yeasts have strong ester production capacities; therefore, they mainly play a role in late fermentation.

### 3.6. Traceability Analysis of Microorganisms

We used Source Tracker to quantitatively analyze the influences of air, Daqu, ground, tools, and water on microbial abundances in the fermented grains at the beginning of fermentation.

Bacteria in the fermented grains from the traditional workshop mainly came from Daqu (83.50%), followed by air (10.68%), and tools (5.67%) (Figure 7a), while fungi mainly came from Daqu (71.49%), followed by air (28.34%) (Figure 7b). In the mechanized workshop fermented grains, 94.04% of the bacterial population came from Daqu, and the air and tools contributed 2.54% and 2.21%, respectively (Figure 7c). The fungal population was mainly derived from Daqu (79.70%) and air (20.19%), while the other sources contributed less than 1% (Figure 7d). In general, the contribution of air in the traditional workshop to bacterial and fungal growth was greater than that in the mechanized workshop, but the bacterial and fungal populations of the grains from both workshops mainly originated from Daqu.

### 3.7. Correlation Analysis of Microorganisms and Key Volatile Compounds

The production of volatile metabolites in fermented grains is closely associated with microbial activity. Therefore, microorganisms with a VIP > 1 were considered differential microorganisms. Using the Mantel test and CCA, the relationship between differential microbial genera and volatile metabolites in fermented grains from the traditional and mechanized workshops was analyzed.

In the traditional workshop, there were five pairs of correlations between bacteria and differential flavor substances and two pairs of correlations between fungi and differential flavor substances. Bacteria showed extremely significant correlations with ethyl acetate and 1-nonyl alcohol among the differential flavor substances (*p* < 0.01), whereas fungi only showed significant correlations with 1-nonyl alcohol (*p* < 0.01) (Figure 8a). In the fermented grains from the mechanized workshop, there were six correlations between bacteria and differential flavor substances and four with fungi. There were extremely significant correlations between bacteria and ethyl acetate and hexanoic acid among the differential flavor substances (*p* < 0.01), whereas no significant correlations were observed between fungi and the differential flavor substances (Figure 8b). Therefore, in the fermented grains from both the traditional and mechanized workshops, the correlation between bacteria and differential flavor substances was more significant than that between fungi and differential flavor substances.

The correlations between differential microorganisms and differential metabolites in fermented grains from the traditional and mechanized workshops were similar. Among the bacteria, only *Lactobacillus* was positively correlated with acids and esters (Figure 9a,b), which may be due to the high relative abundance of *Lactobacillus* in fermented grains. Acetic and hexanoic acid, which are produced by *Lactobacillus* metabolism, provide synthetic precursors for many flavor substances, such as ethyl acetate and ethyl caproate, and inhibit the metabolic function of other bacteria. Among the fungi, *Issatchenkia*, *Kazachstania*, and *Naumovozyma* were positively correlated with acids and esters, indicating that yeasts are the main fungal group in the fermentation process. However, *Lactobacillus*, *Issatchenkia*, *Kazachstania*, and *Naumovozyma* were negatively correlated with 2,4-di-tert-butylphenol in fermented grains from the traditional workshop (Figure 9a), but they were positively correlated in fermented grains from the mechanized workshop (Figure 9b).

From the correlation network diagram of dominant bacteria and fungi with a relative abundance greater than 1% and differential metabolites in the fermented grains from the traditional and mechanized workshop, it can be seen that the correlation between microorganisms and differential metabolites in the fermented grains from the mechanized workshop is stronger than that of those from the traditional workshop (Figure 9c,d).

The correlations between the dominant bacterial genera in the fermented samples were further analyzed using the Pearson algorithm. In both the traditional and mechanized workshops’ fermented grains, *Lactobacillus*, among the top ten dominant bacterial genera screened, was negatively correlated with the other nine bacterial genera (Figure A5). This negative correlation was extremely significant in the grains from the mechanized workshop (*p* < 0.01). This indicated that *Lactobacillus* and other dominant bacterial genera showed an antagonistic relationship, which is consistent with the results of the correlation heat map.

The correlation between differential flavor substances and microorganisms shown by the CCA results is also consistent with the heat map. Only *Lactobacillus* in the bacterial populations of the fermented grains from the traditional and mechanized workshops was positively correlated with differential flavor substances, and these differential flavor substances in grains from the traditional and mechanized workshops were mainly formed during the late fermentation period (Figure 9).

## 4. Conclusions

In this study, high-throughput sequencing technology combined with headspace solid-phase microextraction and GC-MS were used to analyze the microbial community structure and volatile metabolite contents of fermented grains from traditional and mechanized workshops and to reveal the relationships between differential microorganisms and volatile metabolites at the genus level. A total of 71 volatile compounds were detected during the fermentation of the grains using GC-MS. The dominant compounds were esters and alcohols, and ethyl hexanoate, 1-nonyl alcohol, ethyl acetate, acetic acid, butyric acid, furfuryl alcohol, hexanoic acid, and 2,4-di-tert-butylphenol were the differential metabolites between the mechanized and traditionally fermented grain samples. The results of high-throughput sequencing showed that the abundance and diversity of bacteria and fungi in the fermented grains from the traditional workshop were higher than those in the grains from the mechanized workshop. The difference in the microbial communities at the end of fermentation between the grains from the two different types of workshops was more significant than that at the beginning of fermentation, and the bacterial population had a more significant succession rule than that of the fungi throughout the fermentation process. In the fermented grains from both workshops, the dominant bacterial phyla were Firmicutes, Actinobacteria, and Proteobacteria. The dominant bacterial genera were *Lactobacillus*, *Saccharopolyspora*, *Bacillus*, *Acetobacter*, *Weissella*, and *Thermoactinomyces*. The dominant fungal phyla were Ascomycota and Mucoromycota, and the dominant fungal genera were *Saccharomycopsis*, *Issatchenkia*, *Kazachstania*, *Thermoascus*, *Pichia*, and *Rhizopus*. Correlation analysis results showed that in the grains from both the traditional and mechanized workshops, the correlation between bacteria and differential flavor substances was more significant than that of fungi. However, because *Lactobacillus* was negatively correlated with the other nine dominant bacterial genera, only *Lactobacillus* in the bacterial population was positively correlated with differential flavor substances. This study provides a basis for further understanding of the brewing process of Baijiu from the perspective of environmental microorganisms, and also suggests that the existing mechanized brewing method should achieve intelligence and high yield while creating a similar microbial environment to the traditional brewing method as much as possible; “take the essence and eliminate the dross”, so as to improve the quality of Baijiu and promote the development of the Baijiu industry.

## Figures and Tables

**Figure 1 foods-13-03710-f001:**
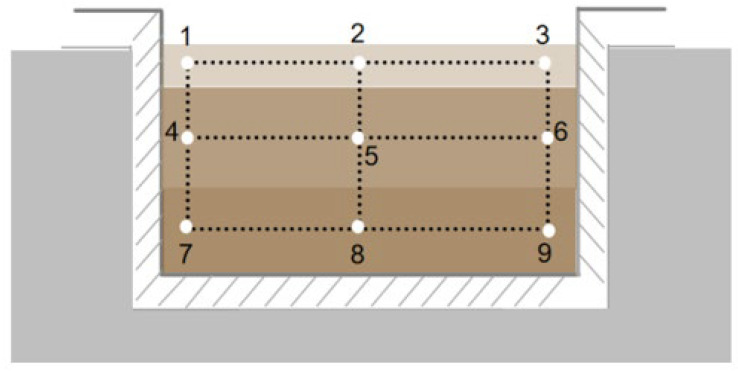
Schematic diagram of the sampling points. The numbers in the figure represent the sampling points, the diagonal filling part represents the pit mud, and the different filling colors in the pit represent the different water content at different levels.

**Figure 2 foods-13-03710-f002:**
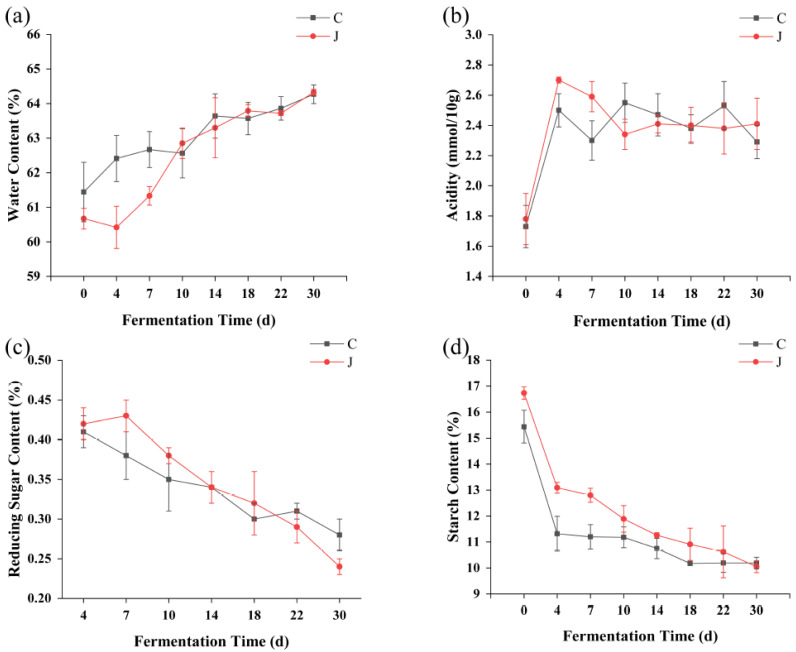
Changes in physicochemical indices during grain fermentation. (**a**) Moisture. (**b**) Acidity. (**c**) Reducing sugar content. (**d**) Starch content. C represents the traditional workshop; J represents the mechanized workshop.

**Figure 3 foods-13-03710-f003:**
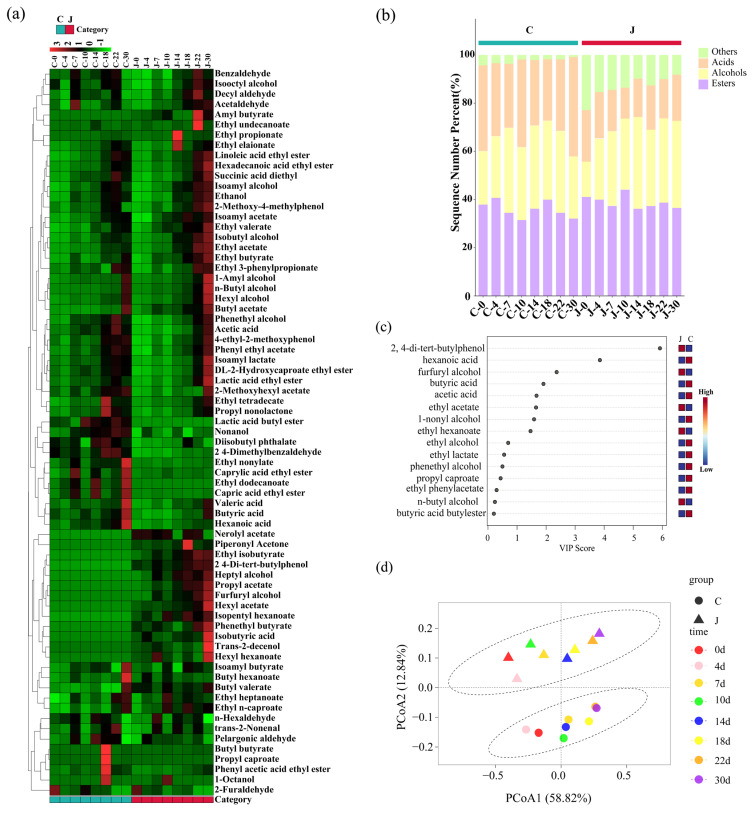
Analysis of volatile flavor compounds content in fermented grains. (**a**) Cluster heat map. (**b**) Histogram of relative content accumulation. (**c**) VIP score map. (**d**) PCoA map. C represents the traditional workshop; J represents the mechanized workshop.

**Figure 4 foods-13-03710-f004:**
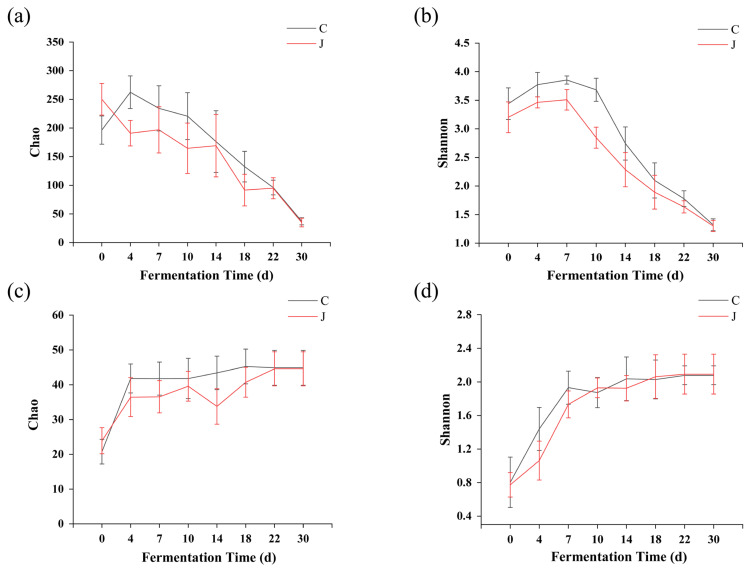
Analysis of microbial α diversity indices during grain fermentation. (**a**) Chao indices of bacteria. (**b**) Shannon indices of bacteria. (**c**) Chao indices of fungi. (**d**) Shannon indices of fungi. C represents the traditional workshop; J represents the mechanized workshop.

**Figure 5 foods-13-03710-f005:**
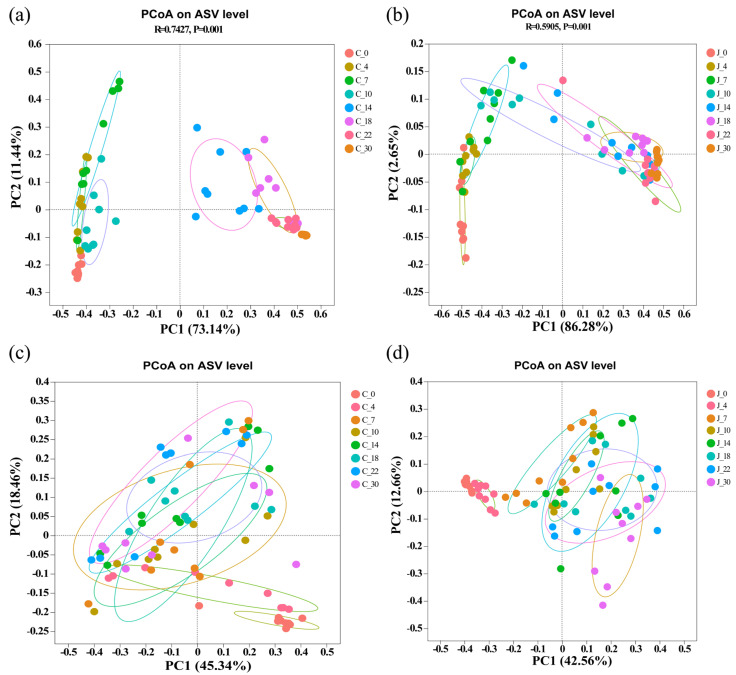
PCoA analysis of microbial communities. (**a**) Bacteria in fermented grains from the traditional workshop. (**b**) Bacteria in fermented grains from the mechanized workshop. (**c**) Fungi in fermented grains from the traditional workshop. (**d**) Fungi in fermented grains from the mechanized workshop. C represents the traditional workshop; J represents the mechanized workshop.

**Figure 6 foods-13-03710-f006:**
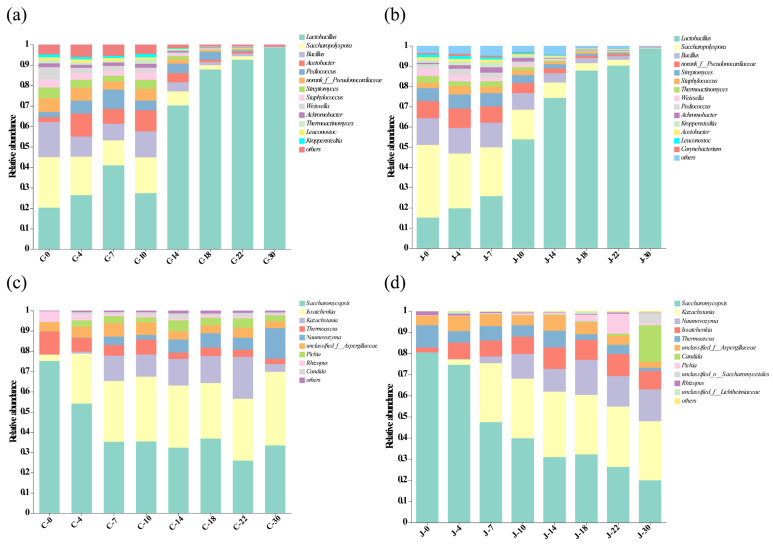
Analysis of microbial community composition at the genus level in fermented grains. (**a**) Bacteria in the fermented grains of the traditional workshop. (**b**) Bacteria in the fermented grains of the mechanized workshop. (**c**) Fungi in the fermented grains of the traditional workshop. (**d**) Fungi in the fermented grains of the mechanized workshop. C represents the traditional workshop; J represents the mechanized workshop.

**Figure 7 foods-13-03710-f007:**
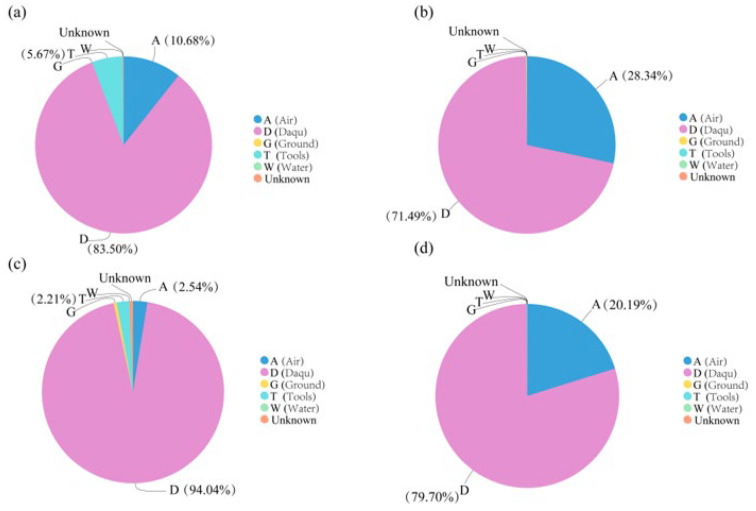
Traceability analysis of microorganisms in fermented grains. (**a**) Bacteria in the fermented grains of the traditional workshop. (**b**) Fungi in the fermented grains of the traditional workshop. (**c**) Bacteria in the fermented grains of the mechanized workshop. (**d**) Fungi in the fermented grains of the mechanized workshop.

**Figure 8 foods-13-03710-f008:**
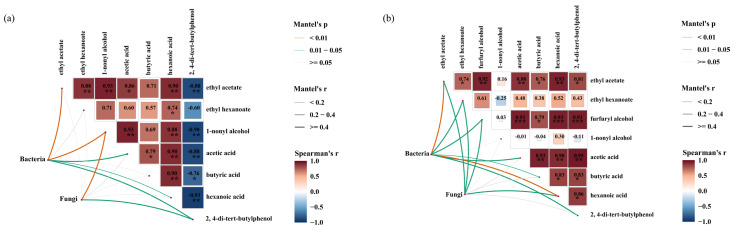
Mantel test analysis of microorganisms and differential flavor substances in fermented grains. (**a**) Traditional workshop. (**b**) Mechanized workshop. "*" means *p* < 0.05, "**" means *p* < 0.01, and "***" means *p* < 0.001.

**Figure 9 foods-13-03710-f009:**
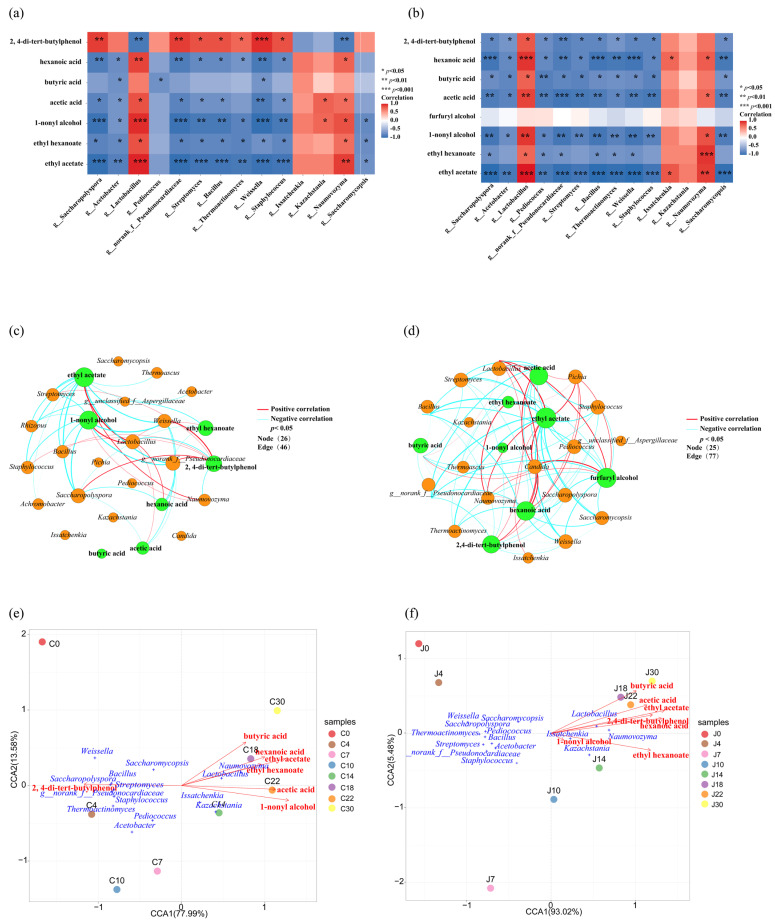
Correlation analysis of microorganisms and differential metabolites in fermented grains. (**a**) Heat maps of the correlation between differential microorganisms and differential flavor substances in the fermented grains from the traditional workshop. (**b**) Heat maps of the correlation between differential microorganisms and differential flavor substances in the fermented grains from the mechanized workshops. (**c**) Network maps of the correlation between the dominant microorganisms and differential flavor substances in the fermented grains from the traditional workshop. (**d**) Network maps of the correlation between the dominant microorganisms and differential flavor substances in fermented grains from the mechanized workshop. (**e**) CCA (canonical correspondence analysis) of microbes and metabolites in the fermented grains from the traditional workshop. The blue letters represent the names of the microbial genus, and the red letters represent the names of the flavor substance. (**f**) CCA (canonical correspondence analysis) of microbes and metabolites in the fermented grains from the mechanized workshop. C represents the traditional workshop; J represents the mechanized workshop.

## Data Availability

The original contributions presented in the study are included in the article, further inquiries can be directed to the author.

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
