# Peer review of "The Impact of Mechanized and Traditional Processes on Microbial Diversity and Volatile Flavor Compound Formation During Xifeng Baijiu Fermentation"

_foods, 2024, doi:10.3390/foods13223710_

Round 1
Reviewer 1 Report (Previous Reviewer 2)
Comments and Suggestions for Authors
The authors have successfully responded to the reviewer's comments and significantly improved the manuscript "The Impact of Mechanized and Traditional Processes on Microbial Diversity and Volatile Flavor Compounds Formation during Xifeng Baijiu Fermentation".
Reviewer 2 Report (Previous Reviewer 1)
Comments and Suggestions for Authors
A second review was carried out on the requested manuscript, and it was observed that the authors addressed all the recommendations derived from the first review. There are no new recommendations for the document.
This manuscript is a resubmission of an earlier submission. The following is a list of the peer review reports and author responses from that submission.
Round 1
Reviewer 1 Report
Comments and Suggestions for Authors
Dear authors, the revised manuscript is interesting. However, the following is suggested:
Line 101: rewrite… 2.1. Experimental Design and Sample Collection
Line 111: rewrite… 2.2. Determination of Physical and Chemical Indices
Line 111: Is any published procedure or method followed for the evaluations of this section? If so, indicate the reference.
Line 119: rewrite… 2.3. DNA Extraction
Line 119: Is any published procedure or method followed for the evaluations of this section? If so, indicate the reference.
Line 125: rewrite… 2.4. PCR Amplification and High-Throughput Sequencing
Line 134: rewrite… 2.5. High-Throughput Sequencing Data Analysis
Line 143: rewrite… 2.6. Determination of Volatile Flavor Substances
Line 143: Is any published procedure or method followed for the evaluations of this section? If so, indicate the reference.
Line 144: insert equipment information, ultrasound (model, brand, country)
Line 145: insert equipment information, centrifuge (model, brand, country)
Line 175: rewrite… 3.1. Changes in Physicochemical Indexes during Fermentation
Line 186: rewrite… 0–4
Line 204: rewrite… 3.2. Changes in Volatile Flavor Compound Contents during Fermentation
Line 227: rewrite… 3.3. Changes in Volatile Flavor compound contents during Fermentation
Note: Subtopics 3.2. and 3.3 are titled the same; is that correct?
Line 248: rewrite… 3.4. Analysis of Microbial Diversity during the Fermentation of Grains
Line 264: Use the same text format as Figure 2
Line 282-286: Could you use some reference for this information?
Line 325: rewrite… 3.5. Succession Trends of Microbial Communities
Line 384: rewrite… 3.6. Traceability Analysis of Microorganisms
Line 402: rewrite… 3.7. Correlation Analysis of Microorganisms and Key Volatile Compounds
Note: Check the correct text format for each of the references. You can review the examples of the Microsoft Word Template described in the authors' guide.
Author Response
Comments 1: rewrite…2.1. Experimental Design and Sample Collection
Response 1: Thank you for pointing this out. I agree with this comment. Therefore, I have corrected the case error in "Experimental design and sample collection" as follows:
Experimental Design and Sample Collection (Page 3, Line 106)
Comments 2: rewrite…2.2. Determination of Physical and Chemical Indices
Response 2: Thank you for pointing this out. I agree with this comment. Therefore, I have corrected the case error in "Determination of physical and chemical indices" as follows:
Determination of Physical and Chemical Indices (Page 3, Line 130)
Comments 3: Is any published procedure or method followed for the evaluations of this section? If so, indicate the reference.
Response 3: Thanks for the reviewer's instructive suggestions. We have now added reference [19] to this section. (Page 4, Line 138)
[19] Deng, L.J.; Zou, X.Y.; Xiong, L.J.; Tang, N. Application of near-infrared spectrometer in the detection of fermented grains of Nongxiang-Jiangxiang Baijiu. Liquor-Making Science & Technology, 2022, 05, 128-132.
Comments 4: rewrite…2.3. DNA Extraction
Response 4: Thank you for pointing this out. I agree with this comment. Therefore, I have corrected the case error in "DNA extraction" as follows:
DNA Extraction (Page 4, Line 139)
Comments 5: Is any published procedure or method followed for the evaluations of this section? If so, indicate the reference.
Response 5: Thanks for the reviewer's valuable advice. We have now added reference [20] to this section. (Page 4, Line 145)
[20] Wang, L.Q.; Tang, P.; Zhao, Q.; Shan, Q.M.G.; Qin, L.Q.; Xiao, D.G.; Li, C.W.; Lu, J.; Guo, X.W. Difference between traditional brewing technology and mechanized production technology of jiangxiangxing baijiu: Micro ecology of zaopei, physicochemical factors and volatile composition. Food Research International, 2024, 192, 114748.
Comments 6: rewrite…2.4. PCR Amplification and High-Throughput Sequencing
Response 6: Thank you for pointing this out. I agree with this comment. Therefore, I have corrected the case error in "PCR amplification and high-throughput sequencing" as follows:
PCR Amplification and High-Throughput Sequencing (Page 4, Line 146)
Comments 7: rewrite…2.5. High-Throughput Sequencing Data Analysis
Response 7: Thank you for pointing this out. I agree with this comment. Therefore, I have corrected the case error in "High-throughput sequencing data analysis" as follows:
High-Throughput Sequencing Data Analysis (Page 4, Line 155)
Comments 8: rewrite…2.6. Determination of Volatile Flavor Substances
Response 8: Thank you for pointing this out. I agree with this comment. Therefore, I have corrected the case error in "Determination of volatile flavor substances" as follows:
Determination of Volatile Flavor Substances (Page 4, Line 164)
Comments 9: Is any published procedure or method followed for the evaluations of this section? If so, indicate the reference.
Response 9: Thanks for the reviewer's valuable advice. We have now added reference [9] to this section. (Page 4, Line 179)
[9] Liu, L.L.; Yang, H.; Jin, X.; Zhang, Y.F.; Yan, Z.K.; Qi, Y.H.; Xu, C. Influence of different wine storage containers on flavor substances of Feng-flavor liquor. Food Science, 2022, 43, 285-293.
Comments 10: insert equipment information, ultrasound (model, brand, country)
Response 10: Thanks for the reviewer very much. According to the reviewer's suggestion, we have added the information of ultrasound. (Page 4, Line 166)
Comments 11: insert equipment information, centrifuge (model, brand, country)
Response 11: Thanks for the reviewer very much. Because we use a refrigerated centrifuge, we do not need a separate refrigerated refrigerator.
Comments 12: rewrite…3.1. Changes in Physicochemical Indexes during Fermentation
Response 12: Thank you for pointing this out. I agree with this comment. Therefore, I have corrected the case error in "Changes in physicochemical indexes during fermentation" as follows:
Changes in Physicochemical Indexes during Fermentation. (Page 5, Line 198)
Comments 13: rewrite… 0–4
Response 13: Thank you for pointing this out. I agree with this comment. Therefore, I have replaced "days 0-4" with "days 0 to 4". (Page 6, Line 213)
Comments 14: rewrite…3.2. Changes in Volatile Flavor Compound Contents during Fermentation
Response 14: Thanks for the reviewer very much. I agree with this comment. Therefore, I have corrected the case error in "Changes in volatile flavor compound contents during fermentation" as follows:
Changes in Volatile Flavor Compound Contents during Fermentation (Page 6, Line 229)
Comments 15: rewrite…3.3. Changes in Volatile Flavor compound contents during Fermentation Note: Subtopics 3.2. and 3.3 are titled the same; is that correct?
Response 15: Thank you for pointing this out. We are very sorry for our wrong writing for the heading. Therefore, I have replaced "Changes in volatile flavor compound contents during fermentation" with "Screening of Key Metabolites and Volatile Flavor Compounds during Fermentation". (Page 7, Line 256)
Comments 16: rewrite…3.4. Analysis of Microbial Diversity during the Fermentation of Grains
Response 16: Thank you for pointing this out. I agree with this comment. Therefore, I have corrected the case error in "Analysis of microbial diversity during the fermentation of grains" as follows:
Analysis of Microbial Diversity during the Fermentation of Grains (Page 7, Line 278)
Comments 17: Use the same text format as Figure 2
Response 17: Thanks for the reviewer's comment very much. According to the reviewer's suggestion, we have changed all the diagrams to the same text format as Figure 2. (Page 8, Line 286-288)
Comments 18: Could you use some reference for this information?
Response 18: Thanks for the reviewer's comment very much. We have now added reference [28] to this information. (Page 9, Line 317)
[28] Wang, X.S. Microbial community structure and microbial source tracking of Chinese light-flavor liquor fermentation in different environments. Ph. D. Dissertation, Jiangnan University, Wuxi, 2018.
Comments 19: rewrite…3.5. Succession Trends of Microbial Communities
Response 19: Thank you for pointing this out. I agree with this comment. Therefore, I have corrected the case error in "Succession trends of microbial communities" as follows:
Succession Trends of Microbial Communities (Page 10, Line 360)
Comments 20: rewrite…3.6. Traceability Analysis of Microorganisms
Response 20: Thank you for pointing this out. I agree with this comment. Therefore, I have corrected the case error in "Experimental design and sample collection" as follows:
Experimental Design and Sample Collection (Page 12, Line 423)
Comments 21: rewrite…3.7. Correlation Analysis of Microorganisms and Key Volatile Compounds. Note: Check the correct text format for each of the references. You can review the examples of the Microsoft Word Template described in the authors' guide.
Response 21: Thank you for pointing this out. I agree with this comment. Therefore, I have corrected the case error in "Correlation analysis of microorganisms and key volatile compounds" as follows:
Correlation Analysis of Microorganisms and Key Volatile Compounds (Page 12, Line 442)
As for the references, we have also modified them according to the template and model essay.

Reviewer 2 Report
Comments and Suggestions for Authors
REVIEW REPORT
TITLE
Title is unclear. Autors should improve the title of manuscript, that is, title should be „The Impact of Mechanized and Traditional Processes on Microbial Diversity and Flavour Compounds Formation during Xifeng Baijiu Fermentation“.
Abstract
Lines 11-12: Sentence is unclear. It should be „This study investigates the fermented grains originated from both mechanized and traditional processes (method) of solid-state fermentation“????? proveri još jednom
INTRODUCTION section
Line 38: „...metabolize....“ Unclear. This word should be deleted.
Line 41: „...populations...“ should be replaced by „...genera...“
Lines 41-43: The first part (from „...The assembly...“ to „...and found that...“) of the sentence is not clear and needs to be corrected.
Lines 57-62: Please provide references for this claim.
Lines 72-76: It would be good to briefly describe the ways of working and the differences between traditional and mechanized Baiju brewing workshops, and/or provide an appropriate reference. Description of traditional and mechanized methods can be done in Supplementary material.
Lines 76-81: Please provide references for this claims.
Line 89: What is the meaning of numbers (906) and (907)?
Generally, many references missed in Introduction section. Autors should cited references that support different sentences in this part of manuscript.
MATERIALS AND METHODS section
Line 102: What is the meaning of numbers (906) and (907)?
Lines 103-105: Please describe in more detail how the samples were taken from the middle and bottom layers.
Line 115-118: Please provide more detailed operational indicators of the FT-IR spectrometer or provide a suitable reference.
RESULTS and DISCUSSION section
Line 177: „...two workshops...“ should be replaced by „...two different types of workshops...“
Line 186: „...two workshops...“ should be replaced by „...two different types of workshops...“ This should be done throughout the manuscript.
Line 189: „Acids are a precursor for the synthesis of Baijiu flavor substances.“ Is it true? Please provide references, if the previous statement is correct.
Line 196-199: Do these differences come from differences in microbial metabolism, or from differences in microflora in the two different types of workshops? The existing wording of the sentence is not precise and requires correction.
Line 210: „Alcohol substances...“ should be replaced by „Alcohols...“
Line 212: „Although the alcohol content...“ should be replaced by „Altough the contents of alcohols...“ Or, in other words, does the word alcohol is synonim for ethanol. In that case, previously mentioned sentence is correct. If alcohol in this sentence was the name of chemical class, which contains various alcohols formed during fermentation, sentence should be corrected.
Figure 3b: The legend on this figure is absent. Please insert it.
Line 219-220: This claim is largely speculative, given that different brewing methods in different types of workshops may lead to different environmental conditions for the development of different species of microorganisms. Therefore, it would be good to describe the brewing procedures in two types of workshops (see the remark for lines 72-76). Otherwise, authors should cite appropriate references.
Lines 282-285: „This may be because, as fermentation progressed, the oxygen content and pH in the pit decreased, and the microenvironment of fermented grains became unfavorable for bacterial growth, while the fungi occupied the bacterial ecological niche and gradually became the predominant fermentation population.“ Different genera and species of microorganisms have different oxygen requirements. In order for the statement to be correct, it is necessary to state (provide references) which of the identified microorganisms are aerobic, anaerobic or facultatively anaerobic.
Lines 288-289: A full stop is missing at the end of the sentence. Also „...were significantly separate“ should be replaced by „...were significantly different.“
Lines 318-324: The explanation given in these lines is what should support the remarks given for the lines 72-76, 196-199, and 219-220.
Lines 338-339: „In addition, the dominant bacterial phylum in feng-flavor and other flavor Baijiu pit mud, such as strong-flavor Baijiu pit mud [16].“ What is the meaning of this sentence? Which bacterial phylum is dominant?
Lines 375-377: „...Saccharomycopsis, the dominant bacterium in grains...“???? Sentence is unclear.
Figure 7: There are no C and J marks in the Figure.
Line 393: (83.0%) is not same as value in Figure 7 (83.50%).
Lines 403-404: What is the meaning of the first sentence in the paragraph?
Lines 409-410: Unclear sentence. It needs to be corrected.
Line 484: ...Mucor..“ should be replaced by „...Mucoromycota“
APENDICES
Figure A. 2. It is clearer if it is written „Venn diagram of ASV of bacteria and fungi in fermented grains in traditional workshop (a, b) and mechanized workshop (c, d)...“ than „Venn diagram of ASV of bacteria and fungi in traditional workshop (a, b) and mechanized workshop (c, d) in fermented grains...“
GENERAL REMARKS
The manuscript is interesting and has a lot of new experimental results that shed a better light on the development and diversity of the microbial community during solid-state fermentation in Xifeng Baijiu production, using two different methods (traditional and mechanized). From that point of view, the manuscript is interesting for publication. However, the authors should correct certain illogicalities and inaccuracies in the text, which have already been presented to them. It is necessary to better describe the selected methods of production in order to gain a clearer insight into the presence of differences in environmental (ecological) factors that would possibly affect the differences in microflora during the production of Xifeng Baijiu using traditional or mechanized methods. This should be supported by appropriate references. Therefore, we believe that the manuscript should be major revisions.
Author Response
Comments 1: Title is unclear. Autors should improve the title of manuscript, that is, title should be ”The Impact of Mechanized and Traditional Processes on Microbial Diversity and Flavour Compounds Formation during Xifeng Baijiu Fermentation“.
Response 1: Thank you for pointing this out. We agree with this comment. Therefore, we have replaced the title of the manuscript with ”The Impact of Mechanized and Traditional Processes on Microbial Diversity and Volatile Flavor Compounds Formation during Xifeng Baijiu Fermentation”. (Page 1, Line 2-4)
Comments 2: Lines 11-12: Sentence is unclear. It should be “This study investigates the fermented grains originated from both mechanized and traditional processes (method) of solid-state fermentation“????? proveri još jednom
Response 2: Thanks for the reviewer's suggestions very much. According to it, we have changed the sentence to “In this study, the differences of physicochemical property, microorganisms, volatile flavor compounds and their correlation in traditional and mechanized processes of Xifeng Baijiu were compared.” (Page 1, Line 12-14)
Comments 3: Line 38: “...metabolize....“ Unclear. This word should be deleted.
Response 3: Thanks for the reviewer's valuable advice. We have deleted the “metabolize” and made the sentence “They can produce amylase”. (Page 1, Line 41)
Comments 4: Line 41: “...populations...“ should be replaced by “...genera...“
Response 4: Thank you for pointing this out. We agree with this comment. So we have replaced “populations” with “genera”. (Page 1, Line 44)
Comments 5: Lines 41-43: The first part (from “...The assembly...“ to “...and found that...“) of the sentence is not clear and needs to be corrected.
Response 5: Thank you for pointing this out. We are very sorry for our no clear writing for this sentence, and we have corrected it as follows:
The assembly and succession of microbial communities in the second round of fermentation of light-flavor Baijiu were studied. It found that Streptomyces, Bacillus, and Lactobacillus were the dominant bacteria in the early stages of fermentation, whereas Lactobacillus was the dominant bacteria in the middle and late stages of fermentation. (Page 1-2, Line 44-48)
Comments 6: Lines 57-62: Please provide references for this claim.
Response 6: Thanks for the reviewer's comment. We have now added references [8] and [9] to this section. (Page 2, Line 65)
[8] Ren, J.M.; Chen, J.P.; Jia, W.; Li, Z.J.; Feng, Y.F.; Li, Y.L. Determination of volatile components in Feng-flavor Baijiu by GC×GC-TOF-MS. China Brewing, 2023, 42, 231-238.
[9] Liu, L.L.; Yang, H.; Jin, X.; Zhang, Y.F.; Yan, Z.K.; Qi, Y.H.; Xu, C. Influence of different wine storage containers on flavor substances of Feng-flavor liquor. Food Science, 2022, 43, 285-293.
Comments 7: Lines 72-76: It would be good to briefly describe the ways of working and the differences between traditional and mechanized Baiju brewing workshops, and/or provide an appropriate reference. Description of traditional and mechanized methods can be done in Supplementary material.
Response 7: Thank you for pointing this out. We are sorry we didn't introduce how do we define two types of workshops, so we have added supplementary content in the part of materials and methods. (Page 3, Line 109-118)
Comments 8: Lines 76-81:Please provide references for this claims.
Response 8: Thanks for the reviewer's comment. We have now added references [13-16] to this section. (Page 2, Line 84,89)
[13]Xu, Y.; Wu, M.Q.; Niu, J.L.; Lin, M.W.; Zhu, H.; Wang, K.; Li, X.T.; Sun, B.G. Characteristics and correlation of the microbial communities and flavor compounds during the first three rounds of fermentation in Chinese sauce-flavor baijiu. Foods, 2023, 12, 207.
[14] Wang, H.; Xi, D.Z.; Huang, Y.G.; Cao, W.T.; You, X.L.; Cheng, P.Y.; Hu, F. Bacterial Community Structure and Diversity in Different Stacking Fermentation Rounds in Mechanized Maotai-Flavor Liquor Brewing. Food Science, 2020, 41, 188-195.
[15] Liu, H.J.; Yu, Y.G.; Wu, Q.; Zhang, X.; Wan, Y.; Xiong, X.; Tan, W.J. Correlation analysis of dominant bacteria and differential metabolites in sauce-flavor Baijiu collected from different fermentation cycles. Food and Fermentation Industries, 2023, 49, 193-198+206.
[16] Yan, P.; Tan, Y.; Li, D.Y.; Shao, J.Y.; Liu, Z.L.; Li, J.; Li, X.Y. Differences in Physicochemical Properties and Microbial Communities of Fermented Grains between Manual and Mechanized Production of Xiaoqu Qingxiang Baijiu. Liquor-Making Science & Technology, 2022, 04, 65-70.
Comments 9: Line 89: What is the meaning of numbers (906) and (907)?
Response 9: Thank you for pointing this out. We are very sorry that we did not clearly point out that these two numbers respectively represent two different workshops, but because their existence has no specific significance, we decided to deleted them.
Comments 10: Line 102: What is the meaning of numbers (906) and (907)?
Response 10: Thank you for pointing this out. We are very sorry that we did not clearly point out that these two numbers respectively represent two different workshops, but because their existence has no specific significance, we also decided to deleted them.
Comments 11: Lines 103-105: Please describe in more detail how the samples were taken from the middle and bottom layers.
Response 11: Thank you for pointing this out. We use a special sampler that can sample at different depths, so we didn't describe in more detail, we have now added tools to the method. (Page 3, Line 120-122)
Comments 12: Line 115-118: Please provide more detailed operational indicators of the FT-IR spectrometer or provide a suitable reference.
Response 12: Thank you for pointing this out. According to the comment, we have now added reference [19] to this section. (Page 4, Line 138)
[19] Deng, L.J.; Zou, X.Y.; Xiong, L.J.; Tang, N. Application of near-infrared spectrometer in the detection of fermented grains of Nongxiang-Jiangxiang Baijiu. Liquor-Making Science & Technology, 2022, 05, 128-132.
Comments 13: Line 177: “...two workshops...“ should be replaced by “...two different types of workshops...“
Response 13: Thank you for pointing this out. According to the reviewer’s suggestion, we have replaced “two workshops” with “two different types of workshops”. (Page 5, Line 200)
Comments 14: Line 186: “...two workshops...“ should be replaced by “...two different types of workshops...“ This should be done throughout the manuscript.
Response 14: Thank you for pointing this out. we have replaced “two workshops” with “two different types of workshops” throughout the manuscript. (Page 6, Line 214)
Comments 15: Line 189: “Acids are a precursor for the synthesis of Baijiu flavor substances.“ Is it true? Please provide references, if the previous statement is correct.
Response 15: Thank you for pointing this out. According to the comment, we have now added reference [21] to this section. (Page 6, Line 218)
[21]Liu, M. Regulation and control of acids in the production process of Nongxiang Baijiu (Liquor). Liquor-Making Science & Technology, 2014, 08, 62-64.
Comments 16: Line 196-199: Do these differences come from differences in microbial metabolism, or from differences in microflora in the two different types of workshops? The existing wording of the sentence is not precise and requires correction.
Response 16: Thank you for pointing this out. In fact, due to the differences of microflora, there are differences in metabolites produced by them, so the root cause is the difference of microflora, we have changed “microbial metabolism”to “microflora”. (Page 6, Line 228)
Comments 17: Line 210: “Alcohol substances...“ should be replaced by “Alcohols...“
Response 17: Thank you for pointing this out. We agree with this comment. So we have replaced “Alcohol substances” with “Alcohols”. (Page 7, Line 240)
Comments 18: Line 212: “Although the alcohol content...“ should be replaced by “Altough the contents of alcohols...“ Or, in other words, does the word alcohol is synonim for ethanol. In that case, previously mentioned sentence is correct. If alcohol in this sentence was the name of chemical class, which contains various alcohols formed during fermentation, sentence should be corrected.
Response 18: Thank you for pointing this out. We are referring to a class of substances in the article, so we have changed this sentence to “Altough the content of alcohols...” (Page 7, Line 242-243)
Comments 19: Figure 3b: The legend on this figure is absent. Please insert it.
Response 19: Thank you for pointing this out. We are sorry we deleted the legend by mistake when we combined the graphics. Now we have added the legend. (Page 6, Line 234)
Comments 20: Line 219-220: This claim is largely speculative, given that different brewing methods in different types of workshops may lead to different environmental conditions for the development of different species of microorganisms. Therefore, it would be good to describe the brewing procedures in two types of workshops (see the remark for lines 72-76). Otherwise, authors should cite appropriate references.
Response 20: Thank you for pointing this out. According to the reviewer's suggestion, we have added a description of the results here, and explain the differences between the two types of workshops that cause such results. (Page 7, Line 250-255)
Comments 21: Lines 282-285: “This may be because, as fermentation progressed, the oxygen content and pH in the pit decreased, and the microenvironment of fermented grains became unfavorable for bacterial growth, while the fungi occupied the bacterial ecological niche and gradually became the predominant fermentation population.“ Different genera and species of microorganisms have different oxygen requirements. In order for the statement to be correct, it is necessary to state (provide references) which of the identified microorganisms are aerobic, anaerobic or facultatively anaerobic.
Response 21: Thank you for pointing this out. We have cited reference [28] in the article to illustrate our conclusions. (Page 8-9, Line 314-317)
[28] Wang, X.S. Microbial community structure and microbial source tracking of Chinese light-flavor liquor fermentation in different environments. Ph. D. Dissertation, Jiangnan University, Wuxi, 2018.
Comments 22: Lines 288-289: A full stop is missing at the end of the sentence. Also “...were significantly separate“ should be replaced by “...were significantly different.“
Response 22: Thank you for pointing this out. According to the reviewer's opinion, we have modified the whole sentence, and the revised sentence is "The types of flavor substances in the fermented grains from the traditional and mechanized workshops were significantly different." (Page 9, Line 319-320)
Comments 23: Lines 318-324: The explanation given in these lines is what should support the remarks given for the lines 72-76, 196-199, and 219-220.
Response 23: Thank you for pointing this out. We agree with the suggestions given by the reviewer, and we also add the description of the two types of workshops in the article, so as to make our discussion based on it.
Comments 24: Lines 338-339: “In addition, the dominant bacterial phylum in feng-flavor and other flavor Baijiu pit mud, such as strong-flavor Baijiu pit mud [16].“ What is the meaning of this sentence? Which bacterial phylum is dominant?
Response 24: Thank you for pointing this out. We are very sorry for our misrepresentation, what we are trying to say here is that these dominant bacteria mentioned above are also the dominant bacteria in the pit mud of feng-flavor Baijiu and some other flavor such as sauce-flavor Baijiu. The corrected sentence in the article is "In addition, these dominant bacteria were also the dominant bacteria in the pit mud during the brewing process of feng-flavor Baijiu and some other flavor such as sauce-flavor Baijiu". (Page 10, Line 374-376)
Comments 25: Lines 375-377: “...Saccharomycopsis, the dominant bacterium in grains...“???? Sentence is unclear.
Response 25: Thank you for pointing this out. We are very sorry for our unclear writing for this. We have changed this sentence to “The relative abundance of the Saccharomycopsis in grains from both types workshops decreased throughout fermentation, from 75.22% to 25.91% and 80.32% to 19.97%, respectively.” (Page 11, Line 414-416)
Comments 26: Figure 7: There are no C and J marks in the Figure.
Response 26: Thank you for pointing this out. We are very sorry for the mistake caused by our carelessness. Now we have deleted these unnecessary contents. (Page 12, Line 438-441)
Comments 27: Line 393: (83.0%) is not same as value in Figure 7 (83.50%).
Response 27: Thank you for pointing this out. We are very sorry for our wrong writing for the figure. We have changed the value in the paper to 83.50%. (Page 12, Line 428)
Comments 28: Lines 403-404: What is the meaning of the first sentence in the paragraph?
Response 28: Thank you for pointing this out. We are very sorry for our mistake, the first sentence of this paragraph should be “The production of volatile metabolites in fermented grains is closely associated with microbial activity”, the previous sentence has been deleted".
(Page 12, Line 443-444)
Comments 29: Lines 409-410: Unclear sentence. It needs to be corrected.
Response 29: Thank you for pointing this out. We have changed this sentence to “In the traditional workshop, there were five pairs of correlations between bacteria and differential flavor substances, and two pairs of correlations between fungi and differential flavor substances.” (Page 12, Line 448-450)
Comments 30: Line 484: ...Mucor..“ should be replaced by „...Mucoromycota“
Response 30: Thank you for pointing this out. We have replaced “Mucor” with“Mucoromycota”. (Page 15, Line 533)
Comments 31: Figure A. 2. It is clearer if it is written “Venn diagram of ASV of bacteria and fungi in fermented grains in traditional workshop (a, b) and mechanized workshop (c, d)...“ than „Venn diagram of ASV of bacteria and fungi in traditional workshop (a, b) and mechanized workshop (c, d) in fermented grains...“
Response 31: Thank you for pointing this out. According to the reviewer's opinion and in order to be consistent with the previous, we modify the figure note as "Venn diagram of ASV of bacteria and fungi. (a) Bacteria in fermented grains of traditional workshop. (b) Fungi in fermented grains of traditional workshop. (c) Bacteria in fermented grains of mechanized workshop. (d) Fungi in fermented grains of mechanized workshop." (Page 16, Line 563-566)

Reviewer 3 Report
Comments and Suggestions for Authors
Dear Authors,
In general, the manuscript is almost excellent; the manuscript is adequate to results presented in this paper the structure of the work is clear and complete. After reading the paper, it is clear that the authors have experience in fermented soy foods and the structure of presented work is clear and looks complete.
The authors of this work used a high-throughput sequencing technology combined with headspace solid-phase microextraction and GC-MS was used to analyze the microbial community structure and volatile metabolite contents of fermented grains from traditional and mechanized workshops and to reveal the relationship between differential microorganisms and volatile metabolites at the genus level.
It is well known, that China's economic development requires an increase in food production potential and this often requires the development of technologies using machines and devices for large-scale industrial production on the one hand, however on the other hand, it is worth paying attention to traditional methods that provide products with higher taste and smell values.
These results have significant implications for the development of Baijiu with certain flavors and directional control of microorganisms during fermentation.
With the progress of science and technology and the development of society, the disadvantages of the traditional brewing process of Chinese Baijiu, including extensive processing, low production efficiency, low standardization, and a low degree of refinement, have been highlighted. Therefore, mechanized Baijiu brewing methods have been developed, and research on the mechanization of Baijiu brewing has gradually increased. Bacillus, Lactobacillus, and Sphingomonas play important roles in the biological regulation of the core microbial communities in grains in different rounds of sauce-flavor Baijiu fermentation.
The authors, studying literature indicate, that the microbial community evolution during manual and mechanized brewing of Xiaoqu Qing-flavor Baijiu showed that Lactobacillus was the most abundant bacterium in both manual and mechanized brewing.
Baijiu is the world's bestselling liquor, with sold over 11 billion liters sold per year, more than all alcohol beverages in the world. Baijiu is not my scope of interest, however, market and trade, as well as, all steps of wines processing chain, cider, and other alcohol beverages coming from distillation, is a subject of my interest. Volatile compounds (e-nose and GC-MS) and color, is a scope of my interest at last research. It is well known, that quality and price of most alcohol are related to many factors, such localization of vineyard, soil, bedrock, climate, that all influenced "teroir". Processing parameters and storage influenced quality as well, and in this poin of view the paper is very important. All of this condition guaranty final quality, brand and price of wine. I known, that Baijiu is related to many factors, and quality and price is very wide. However, Baijiu well known in Asian countries only, should be expressed to readers from western communities. All my remarks are suggestion for future study and will be helpful for other scientist.
I present only few comments below:
1. Lines 220-222 – The figure 3(a) should be separated and presented in vertical form (turn 90o left), that it will be enlarged and more visible. On the other hand all components (flavor compounds) listed below will be they will be arranged on the right side of cluster heat map, lay correctly for reading from left to right, being readable.
2. Lines 262-264 – The figure 4 – Vertical axis concern days, I guess? It should be write days on the end of axis and below you should indicate only number (0, 4, 7, 10, 14, 18, 22, 30). It will be more readable, when days numbers are arranged proportionally to time on the day axis, indicating that at the first stage of grain fermentation bacteria and fungi was analyzed more frequent.
3. Lines 438-439 – The figure 9e and 9f are not readable, even enlarged. The names of microbes and metabolites cover overlap frequently.
4. Lines 490-493 – Conclusions – The Authors concluding should indicate what mechanized procedure allow produce Baijiu more close to the traditional methods instead that realized study provides a theoretical basis for improving the understanding of the Baijiu brewing process from the perspective of environmental microorganisms, which can support the improvement of the quality of Baijiu and promote the advancement of the Baijiu industry.
I also suggested some papers from connected with scope of paper that can improve literature review, where volatile compounds of other fermented products including Baijiu were studied in different condition of storage or at processing: “A systematic, comparative study on the physicochemical properties, volatile compounds, and biological activity of typical fermented soy foods”; “A straightforward, sensitive and reliable strategy for ethyl carbamate detection in the by-products from Baijiu by Enzyme-linked immunosorbent assay”.
I recommend to be in corrected form after minor revision, the paper don’t need any additional improvement in my opinion.
Comments on the Quality of English Language
The manuscript is written in understandable language for reader, however it should be slightly corrected consequently to American or English; eg. In the Title: Flavour or Flavor
Author Response
Comments 1: Lines 220-222 – The figure 3(a) should be separated and presented in vertical form (turn 90 left), that it will be enlarged and more visible. On the other hand all components (flavor compounds) listed below will be they will be arranged on the right side of cluster heat map, lay correctly for reading from left to right, being readable.
Response 1: Thank you for pointing this out. We agree with this comment. Therefore, we have modified the layout of the image, and the modified figure is arranged as follows: (Page 6, Line 234)
Comments 2: Lines 262-264 – The figure 4 – Vertical axis concern days, I guess? It should be write days on the end of axis and below you should indicate only number (0, 4, 7, 10, 14, 18, 22, 30). It will be more readable, when days numbers are arranged proportionally to time on the day axis, indicating that at the first stage of grain fermentation bacteria and fungi was analyzed more frequent.
Response 2: Thank you for pointing this out. We agree with this comment. Therefore, we have modified the figure, and the revised figure is as follows: (Page 8, Line 285)
Figure 4. Analysis of microbial α diversity indices during grain fermentation. (a) Chao indices of bacteria. (b) Shannon indices of bacteria. (c) Chao indices of fungi. (d) Shannon indices of fungi. C represents the traditional workshop; J represents the mechanized workshop.
Comments 3: Lines 438-439 – The figure 9e and 9f are not readable, even enlarged. The names of microbes and metabolites cover overlap frequently.
Response 3: Thank you for pointing this out. According to the reviewer's suggestion, we have made adjustment to the figure 9e and 9f and added explanations in the diagram notes. (Page 14, Line 479)
Figure 9. (e). CCA analysis (canonical correspondence analysis).of microbes and metabolites in fermented grains from traditional workshop. The blue letters represent the names of the microbial genus and the red letters represent the names of the flavor substance. (f). CCA analysis (canonical correspondence analysis).of microbes and metabolites in fermented grains from mechanized workshop. C represents the traditional workshop; J represents the mechanized workshop.
Comments 4: Lines 490-493 –Conclusions – The Authors concluding should indicate what mechanized procedure allow produce Baijiu more close to the traditional methods instead that realized study provides a theoretical basis for improving the understanding of the Baijiu brewing process from the perspective of environmental microorganisms, which can support the improvement of the quality of Baijiu and promote the advancement of the Baijiu industry.
Response 4: Thank you for pointing this out. According to the reviewer's suggestion, we have made some changes at the end of the conclusion, “This study provides a basis for further understanding of the brewing process of Baijiu from the perspective of environmental microorganisms, and also suggests that the existing mechanized brewing method should achieve intelligence and high yield while creating similar microbial environment to the traditional brewing method as much as possible, "take the essence and eliminate the dross", so as to improve the quality of Baijiu and promote the development of Baijiu industry.” (Page 15, Line 540-545)
- Response to Comments on the Quality of English Language
Point 1: The manuscript is written in understandable language for reader, however it should be slightly corrected consequently to American or English; eg. In the Title: Flavour or Flavor
Response 1: Thanks for the reviewer's valuable suggestion, we have corrected the language problems in the article, for example, “flavour” has been changed to “flavor”.
- Additional clarifications
In addition, thanks for the two references recommended by the reviewers, which are very useful to us and we have cited them in our article.
